# *Pseudomonas putida* Metallothionein: Structural Analysis and Implications of Sustainable Heavy Metal Detoxification in Madinah

**DOI:** 10.3390/toxics11100864

**Published:** 2023-10-16

**Authors:** Munazzah Tasleem, Abdel-Aziz A. A. El-Sayed, Wesam M. Hussein, Abdulwahed Alrehaily

**Affiliations:** 1School of Electronic Science and Engineering, University of Electronic Science and Technology of China, Chengdu 610054, China; 2Biology Department, Faculty of Science, Islamic University of Madinah, Madinah 42351, Saudi Arabia; 3Chemistry Department, Faculty of Science, Islamic University of Madinah, Madinah 42351, Saudi Arabia

**Keywords:** *Pseudomonas putida*, in silico bioremediation, heavy-metal docking, post-translational modifications, protein–protein interaction

## Abstract

Heavy metals, specifically cadmium (Cd) and lead (Pb), contaminating water bodies of Madinah (Saudi Arabia), is a significant environmental concern that necessitates prompt action. Madinah is exposed to toxic metals from multiple sources, such as tobacco, fresh and canned foods, and industrial activities. This influx of toxic metals presents potential hazards to both human health and the surrounding environment. The aim of this study is to explore the viability of utilizing metallothionein from *Pseudomonas putida* (*P. putida*) as a method of bioremediation to mitigate the deleterious effects of pollution attributable to Pb and Cd. The use of various computational approaches, such as physicochemical assessments, structural modeling, molecular docking, and protein–protein interaction investigations, has enabled us to successfully identify the exceptional metal-binding properties that metallothionein displays in *P. putida*. The identification of specific amino acid residues, namely GLU30 and GLN21, is crucial in understanding their pivotal role in facilitating the coordination of lead and cadmium. In addition, post-translational modifications present opportunities for augmenting the capacity to bind metals, thereby creating possibilities for focused engineering. The intricate web of interactions among proteins serves to emphasize the protein’s participation in essential cellular mechanisms, thereby emphasizing its potential contributions to detoxification pathways. The present study establishes a strong basis for forthcoming experimental inquiries, offering potential novel approaches in bioremediation to tackle the issue of heavy metal contamination. Metallothionein from *P. putida* presents a highly encouraging potential as a viable remedy for environmental remediation, as it is capable of proficiently alleviating the detrimental consequences related to heavy metal pollution.

## 1. Introduction

The contamination of heavy metals, specifically Cd and Pb, in water bodies is posing significant challenges for the environment in Madinah, Saudi Arabia, [1]. Cd, a well-known industrial and environmental contaminant, infiltrates the city of Madinah through various pathways, including the prevailing use of tobacco, a customary practice in the region. It is observed that Cd is frequently present in fresh and canned fish, and vegetable products, which are commonly consumed as staple foods in the local diet. At the same time, the issue of Pb contamination in Madinah is distinguished by multiple factors, including the employment of lead additives in motor fuels, the presence of lead water pipes, the use of lead-containing paints, and the prevalence of smoking practices. Additionally, canned food and beverages are observed to have emerged as potential sources of Pb exposure for the inhabitants of the city. The local environment in Madinah has been impacted by the extensive dispersion of heavy metals as a result of the swift industrialization in the area. The progression of industrialization persists in molding the modern world, with fresh technologies and trades consistently emerging. Microorganisms exhibit exceptional adaptability and play a crucial role in a vast array of ecological processes. Numerous studies have highlighted the importance of these processes, which include nutrient cycling, primary production, and the degradation of various environmental pollutants. Bioremediation strategies utilizing the metabolic capabilities of microorganisms have gained popularity in recent years as eco-friendly and sustainable methods for reducing heavy metal pollution [2]. These methods utilize the ability of bacteria, fungi, and other microorganisms to accumulate or transform heavy metals into less toxic forms. The use of genetically engineered microbes with enhanced metal tolerance and remediation capabilities represents a cutting-edge avenue of research in this field. Understanding the complex interactions between microorganisms and heavy metals in various environmental contexts has also opened up new avenues for the development of tailored bioremediation strategies. Researchers are investigating the efficacy of microbial consortia, which are synergistic combinations of multiple microbe species, in remediating complex heavy metal contamination scenarios [3]. Prior research has conducted a comprehensive analysis of the groundwater quality in the Madinah region, yielding valuable information regarding the levels of heavy metal contaminants present in this geographical area. The research findings revealed elevated levels of various heavy metals, including arsenic (As), cadmium (Cd), lead (Pb), copper (Cu), iron (Fe), chromium (Cr), nickel (Ni), manganese (Mn), vanadium (V), mercury (Hg), cobalt (Co), and zinc (Zn), within the southern region of Madinah [4,5]. Comprehending the implications of pollution on the ecological and human well-being of the area mandates the examination of natural defense mechanisms, such as metallothionein in microorganisms like *P. putida*. These safeguarding measures could potentially play an important part in limiting the adverse impacts of pollution caused by heavy metals [6,7,8]. Heavy metals, such as Pb and Cd, are widely distributed in the environment, posing risks to the surrounding environment and human health [9]. Toxic metals have the potential to be introduced into the environment through a range of channels, encompassing industrial processes, mining endeavors, agricultural methods, and inadequate waste management procedures. The potential impact of heavy metal contamination on vulnerable groups, specifically children and pregnant women, is a matter of significant concern [10]. Children, specifically, exhibit a heightened vulnerability to the detrimental consequences associated with exposure to lead. Lead exposure can occur through multiple paths, encompassing contact with lead-based paint, contaminated soil, dust, air, and sometimes even food [11]. The exposure of children to lead may lead to noteworthy consequences, comprising impairments in behavior and cognition, reduction in intelligence quotient (IQ), hampered physical development, auditory impairments, and the onset of anemia. Lead ingestion even in minute quantities may lead to serious consequences, such as seizures, comatose state, and fatality. Lead exposure can harm fetal development and increase the likelihood of preterm delivery in pregnant women, which makes them vulnerable [12,13]. Aquatic organisms lack a biological necessity for the presence of Pb, as it is deemed non-essential [14,15]. Long-term exposure to cadmium has been found to have detrimental effects on the growth, reproduction, immunological and endocrine systems, development, and behavior of aquatic creatures [16,17].

*P. putida* is a rod-shaped, Gram-negative bacterium that is frequently found in contaminated water bodies, vegetation, and soil. The adaptability of *P. putida* is clear in its opportunistic utilization of a varied range of nutrients, its speedy rate of growth, and its capacity to endure oxidative stress and exposure to harmful substances. The analysis of *P. putida* and its potential to degrade xenobiotics commenced in the 1960s and has subsequently been a voyage to expose its hereditary, biochemical, and physiological traits [18]. *P. putida’s* demonstrated metal and antibiotic resistance properties make it an intriguing candidate for further exploration, particularly in the context of bioremediation [19,20,21]. These resistance traits can be advantageous for their potential use in environmental bioremediation applications. Advancements in technology have further propelled the study of *P. putida*, enabling systems-level profiling and targeted genetic and genomic modifications. These advances have broadened the prospects for industrial applications, leveraging the bacterium’s biochemical capabilities and the expanding knowledge and technology surrounding it. This study intends to illuminate the genetic and metabolic components of *P. putida’s* bioremediation abilities and create a path for focused efforts to alleviate the detrimental impacts of lead and cadmium contamination.

The deployment of efficient and sustainable techniques for the detoxification of heavy metals remains crucial in addressing the detrimental effects associated with lead and cadmium [22]. *P. putida* is a bacterium that is very frequently employed and has been extensively researched due to its ability to break down and eliminate different pollutants, making it an exceptionally auspicious candidate for removing heavy metals [23,24,25].

This research provides a significant contribution to the expanding field of in silico bioremediation and has the potential to advance ecologically sustainable approaches to address the issue of heavy metal contamination. Through the utilization of in silico methodologies, we scrutinize the underlying mechanisms that enable *P. putida* to effectively eliminate these hazardous metals from polluted environments. By means of computational modeling and bioinformatics tools, our aim is to reveal the subtle interactions between *P. putida* and heavy metals, identify critical pathways and enzymes involved in the bioremediation process, and ultimately contribute to the creation of sustainable strategies for heavy metal detoxification. This study makes a valuable contribution to the expanding domain of in silico bioremediation and offers potential for the advancement of ecologically sustainable strategies to address the issue of heavy metal contamination.

## 2. Material and Methods

### 2.1. Physicochemical Insights: Analysis with EXpasy ProtParam Tool

The protein sequence of metallothionein from *P. putida* strain 15420352, accession ID: QKL07673, retrieved from the National Center for Biotechnology Information (NCBI; https://www.ncbi.nlm.nih.gov/ accessed on 2 February 2023), was subjected to analysis using the EXpasy ProtParam tool (https://web.expasy.org/protparam/, accessed on 2 February 2023). This computational tool computes a range of physicochemical parameters that provide insights into the structure and functional characteristics of the protein. The factors considered in this study encompassed the molecular weight, theoretical isoelectric point (pI), the proportion of negatively and positively charged residues, estimated half-life, instability index (II), aliphatic index, and grand average of hydropathicity (GRAVY) value [26].

### 2.2. Functional Profiling: Unveiling Potential Roles Using VicmPred Algorithm

The functional prediction tool, VicmPred, was utilized to perform a comprehensive study of the metallothionein sequence derived from *P. putida*. The VicmPred (http://www.imtech.res.in/raghava/vicmpred/, accessed on 2 February 2023) algorithm employs pattern-based methodologies for the purpose of predicting functional classifications that are linked to a given protein. Our objective was to elucidate the potential functional roles of metallothionein in relation to its binding capabilities with lead and cadmium by inputting the protein sequence [27].

### 2.3. Structural Insights: Revealing Architecture via Superfamily 1.75 Tool

The protein sequence of metallothionein from *P. putida* was subjected to structural classification using the Superfamily 1.75 tool (https://supfam.org/SUPERFAMILY/ accessed on 2 February 2023). This tool identifies conserved domains and classifies proteins into structural classes, folds, and superfamilies based on their sequence features [28].

### 2.4. Regulatory Prospects: Post-Translational Modifications Explored with MusiteDeep Tool

The protein sequence of metallothionein from *P. putida* was analyzed for post-translational modifications (PTMs) using the MusiteDeep tool (https://www.musite.net/ accessed on 2 February 2023). The bioinformatics tool shown herein utilizes sequence characteristics to identify potential post-translational modification (PTM) sites and assign corresponding scores. This tool provides valuable insights into the potential regulatory alterations that may impact protein functionality [29].

### 2.5. Evolutionary Significance: Functional Annotation through EggNOG 6.0 Database

In this study, the protein sequence of metallothionein derived from *P. putida* (UniProt ID: Q88HU1) was submitted to computational analysis utilizing the EggNOG 6.0 database. EggNOG (http://eggnog6.embl.de/ accessed on 3 February 2023) offers valuable insights pertaining to the functional annotations, domains, and evolutionary relationships of proteins [30].

### 2.6. Network Exploration: Protein–Protein Interactions Mapped via STRING Database

The metallothionein sequence derived from *P. putida* was subjected to computational analysis utilizing the STRING database (https://string-db.org/ accessed on 6 February 2023). The STRING database offers significant insights into protein–protein interactions (PPIs), molecular activities, biological processes, and linked protein domains [31].

### 2.7. Binding Assessment: Capacity Probed using PredictProtein Tool

In order to assess the binding capacity of the metallothionein derived from Pseudomonas putida, we utilized the PredictProtein tool (https://predictprotein.org/ accessed on 7 February 2023), a reliable platform for performing computational protein analysis. This complete tool enables the anticipation of secondary structure, solvent accessibility, topology, disordered regions, relative B-value dynamics, binding interactions, disulfide bond formation, and conservation scores. The methodology employed in this study consisted of loading the protein sequence into the PredictProtein software, and afterward, analyzing and interpreting the predictions given by the program [32].

### 2.8. Protein Structure Prediction and Validation

In this study, we intended to conduct homology modeling of the *P. putida* metallothionein sequence. Four distinct programs, including the PHYRE2 Protein Fold Recognition Server (http://www.sbg.bio.ic.ac.uk/phyre2/ accessed on 9 February 2023), Robetta (http://robetta.bakerlab.org/ accessed on 9 February 2023), ModWeb (https://modbase.compbio.ucsf.edu/modweb/ accessed on 9 February 2023), and SwissModel (https://swissmodel.expasy.org/ accessed on 9 February 2023), were utilized for this purpose. First, the primary amino acid sequence of *P. putida* metallothionein was determined. This sequence was then submitted to the aforementioned four homology modeling tools. The PHYRE2 Protein Fold Recognition Server is a web-based application that predicts protein structure using a combination of profile alignment and secondary structure prediction [33]. Robetta is an additional web-based homology modeling application that utilizes the Rosetta software suite [34]. The ModWeb-ModBase Search Page is a component of the ModBase database, which contains homology models for protein sequences [35]. SwissModel is a popular homology modeling application that employs a template-based methodology [36]. The modeled protein structures were subjected to validation using two distinct tools: ProQ3 (https://proq3.bioinfo.se/pred/ accessed on 15 February 2023) [37] and the Protein Structure Validation Suite (PSVS) (https://montelionelab.chem.rpi.edu/PSVS/psvs2/ accessed on 20 February 2023). These tools were employed to assess the quality and reliability of the generated protein structures.

### 2.9. Molecular Docking of Heavy Metals (Lead and Cadmium) with Metallothionein (MT) Modeled Structure and Intramolecular Interaction Analysis

For the purpose of investigating the binding interactions between heavy metals and the metallothionein (MT) modeled structure, molecular docking was conducted using CDocker, a module available within the Discovery Studio software suite (Dassault Systems, BIOVIA Corp., San Diego, CA, USA, v 21.1). The procedure involved the preparation of the protein structure, ligands, and the definition of a binding sphere around the active site. The ligands employed for the simulations were lead and cadmium, utilizing their respective crystallographic structures. Prior to docking, the ligand structures were optimized through the addition of hydrogen atoms and the adjustment of charges using the tools provided by Discovery Studio. Using the CDocker module of Discovery Studio, the docking of heavy metals and the modeled metallothionein structure was executed. The procedure consisted of several essential steps. The binding site was initially defined using the “Input Site Sphere” parameter, with center coordinates of (x, y, and z: 16.2944, 12.409, and −7.49207) and a radius of 32.941 Å. This specified the region within which the calculations for docking would be performed. For the simulations of docking, a set of parameters is configured. The “Top Hits” parameter was set to 10, indicating that the top 10 binding poses for the heavy metal–ligand would be taken into account. To facilitate analysis, a “Pose Cluster Radius” of 0.1 was applied to group similar poses together. To facilitate the investigation of ligand binding modes, “Random Conformations” were generated for the ligand, with ten conformations considered. The “Dynamics Steps” parameter was subsequently set to 1000 and the “Dynamics Target Temperature” parameter was set to 1000 K to enable dynamic movements during the docking procedure. Enabling “Include Electrostatic Interactions” accounted for electrostatic interactions. This ensured that electrostatic forces played a role in the interactions between the ligand and the receptor during docking. Using the “Orientations to Refine” parameter, ten different orientations of the ligand were refined, while the “Maximum Bad Orientations” parameter set a maximum of 800 orientations that did not meet the energy criteria defined by the “Orientation vdW Energy Threshold” of 300 kcal/mol. This refined the selection process for orientation. During docking, the “Simulated Annealing” method was utilized, with “Heating Steps” set to 2000 and “Heating Target Temperature” set to 700 K to facilitate a gradual increase in temperature. To stabilize the binding poses, “Cooling Steps” were set to 5000, and the “Cooling Target Temperature” was set to 300 K. Advanced settings included the use of the CHARMM forcefield and selection of the “Ligand Partial Charge Method” as Momany-Rone. To control the amount of potential energy used during the calculations, the “Use Full Potential” parameter was set to False. Following the docking simulations, the resulting ligand–receptor complexes were subjected to final energy minimization using the full potential with a gradient tolerance of 0 for precision. A grid extension of 8 Å was applied around the binding site of the receptor to account for the flexibility of the ligand binding. The docked complexes were evaluated for close intramolecular interactions.

## 3. Results

### 3.1. Sequence Analysis

The determination of the molecular weight of metallothionein from *P. putida* yielded a value of approximately 7907.74 Da. The compact dimensions of this size offer notable benefits in the context of bioremediation applications [26]. The isoelectric point (pI) of the protein is theoretically calculated to be 4.90. This characteristic offers valuable information regarding the distribution of charge inside the protein [38]. The MT consists of 12 residues that possess a negative charge (Asp and Glu), whereas 7 residues have a positive charge (Arg and Lys) [39]. Negatively charged residues may increase the protein’s ability to sequester the heavy metals: Pb and Cd [40,41]. The MT’s half-life is estimated to be around 30 h. The instability index, as determined through the calculations, is 51.92. A protein exhibiting a value beyond 40 implies a potential for instability under in vitro conditions. Nevertheless, it is important to acknowledge that the in vivo settings may exhibit variations as a result of the defensive mechanisms present within the cellular environment [42]. The protein’s thermostability can be inferred from its aliphatic index, which has been calculated to be 35.81. The significance of this characteristic lies in its ability to maintain the protein’s functionality throughout a range of environmental circumstances [43]. The protein exhibits a primarily hydrophilic character, as evidenced by its grand average hydropathicity (GRAVY) score of −0.695. The property of being hydrophilic could potentially enhance the ability to engage in interactions with both aqueous environments and metal ions [44].

The VicmPred analysis revealed distinctive functional class scores for the metallothionein from *P. putida*. These scores reflect the protein’s potential roles in various biological functions, shedding light on its multifaceted nature. Cellular Process (Score: 0.92678171): The high score suggests the involvement of metallothionein in critical cellular processes. Information Molecule (Score: −0.13593004): The negative score implies a lower association with information-related functions. Metabolism (Score: 0.99999996): The nearly perfect score highlights the potential participation of metallothionein in metabolic processes. Virulence Factors (Score: −0.57314452): The negative score suggests a relatively lower likelihood of the metallothionein’s involvement in virulence-related functions. This aligns with its potential role in detoxification rather than pathogenicity.

The protein is categorized as a member of the “Small proteins” class, which denotes its somewhat condensed size. This property is consistent with the concept that metallothionein often possess a compact size, which facilitates their effective coordination of metal ions within their constrained structural environment [45]. The protein falls within the structural category known as the “Metallothionein” fold, which aligns with its established role in the binding and sequestration of metal ions. The structural motif known as the “Metallothionein” fold is distinguished by the existence of conserved cysteine residues that play a crucial role in coordinating metal ions. The protein is categorized under the superfamily known as “Metallothionein” based on its categorization [46].

The in silico PTM analysis using MusiteDeep unveiled various modifications in the metallothionein from *P. putida*. Position 2 (N): Asparagine at position 2 showed a low N-linked glycosylation score of 0.034, which may influence the protein’s structure and function [47]. Position 4 (Q): Glutamine at position 4 exhibited a significant pyrrolidone carboxylic acid score of 0.543. This modification could contribute to conformational changes affecting metal ion coordination. Positions 6, 8, 11, 13, 42, 47, and 49 (C): Cysteine residues at various positions displayed notable S-palmitoyl cysteine scores, indicating potential palmitoylation events. Palmitoylation could influence the protein’s subcellular localization and interactions [48,49]. Positions 9 and 14 (T): Threonine residues at positions 9 and 14 exhibited moderate phosphothreonine scores, suggesting possible phosphorylation sites that may modulate metal ion binding. Positions 25, 38, and 52 (K): Lysine residues at specific positions displayed significant scores for various PTMs, including ubiquitination and acetylation [50]. Positions 34, 56, 65, 69, and 73 (S): Serine residues at multiple positions demonstrated varying phosphorylation and glycosylation scores. Phosphorylation and glycosylation could affect metal coordination and the protein’s affinity for lead and cadmium [51]. Positions 41, 53, 67, 71, and 74 (P): Proline residues at different positions exhibited substantial hydroxyproline scores, indicating possible hydroxylation events that could influence protein structure, as shown in Figure 1.

### 3.2. Ortholog Identification and Analysis of P. putida Metallothionein

The categorization and putative functional annotations within the protein family of metallothionein from *P. putida* were determined by the EggNOG analysis. The existence of conserved domains associated with metal binding provides more evidence for its involvement in the processes of metal detoxification and bioremediation, as presented in Figure 2.

### 3.3. Protein Interaction Profiling of P. putida Metallothionein

The utilization of the STRING database for the in silico examination of the metallothionein sequence derived from *P. putida* yielded significant findings pertaining to the protein’s interactions, functions, and putative molecular pathways linked to its involvement in metal binding and bioremediation. The study yielded a total of 11 nodes, which represent discrete proteins or protein components that may be involved in interactions with metallothionein. The mentioned nodes serve as representations of prospective partners or molecules that have the capacity to participate in activities linked with metallothionein. The network displays a total of 15 edges, which symbolize the various connections or interactions among the nodes. The edges depicted in the diagram represent the plausible physical or functional connections between metallothionein and other proteins, within the framework of its biological functions. The mean node degree of 2.73 signifies that, on average, every node (representing a protein) inside the network is linked to around 2.73 additional nodes. This metric offers valuable insights into the degree of interaction complexity and potential functional associations between metallothionein and other proteins. The metallothionein network displays a significant degree of clustering, which is substantiated by its mean local clustering coefficient of 0.889. This suggests that proteins affiliated with metallothionein are more prone to forming interconnected relationships with one another. This observation indicates the existence of functionally interconnected protein clusters inside the network. The anticipated count of edges, approximated to be 10, functions as a benchmark for evaluating the network’s relevance. When the count of edges observed in a network far exceeds the anticipated quantity, it signifies the existence of significant interactions within the system. The *p*-value of PPI enrichment, which stands at 0.0968, serves as a quantitative measure of the statistical significance that is linked to the protein–protein interactions that have been observed. The molecular function of DNA binding implies that metallothionein may have a role in DNA interaction, potentially contributing to gene regulation or other activities related to DNA. The identified protein domains and features imply that metallothionein possesses several distinct structural and functional characteristics: MerR Transcriptional Regulator, MerR HTH Domain Type, and Superfamily of Putative DNA-Binding Domains, as shown in Figure 3.

### 3.4. Secondary Structure and Sequence Analysis of P. putida: Insights into Structural Characteristics and Functional Implications

The insightful PredictProtein analysis brought to light the occurrence of not one but two sublime alpha-helices (Residues: 28–33, and 58–63), and a solitary beta-strand (Residues: 27) adorning the metallothionein sequence. These secondary structure elements contribute to the protein’s overall fold and potential metal-binding sites. The analysis revealed that numerous residues, specifically residues 6, 8, 11, 13, 28, 29, 32, 33, 47, 58, and 62, are evidently exposed to the solvent. This suggests that these residues could potentially be involved in various interactions, such as metal binding. Residues with high B-values (1–3, 23, 51–56, 71–74) suggest increased flexibility in these regions. This flexibility might be crucial for accommodating bound metal ions and undergoing conformational changes upon metal binding. Several significant regions that are conserved have been detected, which are located at residues 1, 6–8, 11, 13, 27–29, 31–33, 35, 36, 39, 42, 47, 49, 58, 59, 61, 64, 65, and 67–73, as shown in Figure 4. These conserved regions may signify critical functional sites involved in metal coordination and protein stability.

### 3.5. Three-Dimensional Structure Prediction and Validation

The metallothionein sequence from *P. putida* was modeled using four distinct homology modeling techniques. The validity of the predicted structures derived from PHYRE2, Robetta, ModWeb, and SwissModel was evaluated and compared. The degree of structural similarity among the models produced by PHYRE2, Robetta, ModWeb, and SwissModel varied. The overall fold and secondary structure elements of the models were generally consistent. However, there were variations in the loop regions and local structural characteristics, as shown in Figure 5.

The protein structures generated by Phyre2 and Robetta covered the complete range of residues, spanning 1 to 74. On the other hand, both ModWeb and SwissModel generated structures comprising residues 1 to 73. The ProQ3 scores, which reflect the predicted global model quality, reveal differences among the computational methods. Robetta achieved the highest ProQ3 score of 0.467, indicating greater confidence in its structural predictions. Phyre2 followed with a score of 0.383, while ModWeb recorded a ProQ3 score of 0.000, raising concerns about the reliability of its model. The Ramachandran plot analysis evaluated the backbone torsion angles’ conformations, highlighting their agreement with preferred regions of the plot. Phyre2: 84.1% of residues fall within the most favored region, with 12.7% in additionally allowed regions and 3.2% in generously allowed regions. Robetta outperformed other methods, with 95.4% of residues in the most favored region, showcasing its superior prediction accuracy. ModWeb and SwissModel attained 87.1% and 75.8% in the most favored region, respectively. Close contacts and deviations from the ideal geometry provide insights into potential steric clashes and structural distortions. All methods demonstrated no close contacts within a 2.2 Å distance, suggesting the absence of severe steric clashes. Robetta and SwissModel displayed the lowest RMS deviations for bond angles and bond lengths (2.2° and 0.016 Å, respectively), indicating better agreement with the ideal geometries. The global quality scores offer a comprehensive assessment of various aspects of the protein models. The Verify3D scores were lowest for Robetta (−5.78) and highest for ModWeb (−6.90), suggesting reasonable agreement between the models and the expected 3D protein structures. The ProsaII scores were negative for all methods, with Robetta, Phyre2, and SwissModel exhibiting relatively close values, while ModWeb had a positive score, possibly indicating deviations from the native-like conformations. Procheck analysis highlighted variations in the phi-psi angles and overall structure quality. Robetta displayed favorable results, while ModWeb and SwissModel exhibited negative scores, suggesting potential issues with their phi-psi angles. The MolProbity Clash-score indicates the presence of steric clashes, with lower scores indicating fewer clashes. Phyre2 displayed a substantially negative Clash-score (−19.42), suggesting minimal steric clashes. Robetta, ModWeb, and SwissModel all had positive Clash-scores, implying varying degrees of steric clashes (Table 1).

### 3.6. Molecular Docking and Intramolecular Interactions

The molecular docking simulations provided insights into the binding characteristics of heavy metals lead and cadmium within the metallothionein (MT) modeled structure’s active site. In the case of lead binding, the docking revealed a CDocker energy of −0.10587, indicating a favorable binding affinity. This was corroborated by a corresponding CDocker Interaction energy of −0.10587, highlighting the stability of the lead–MT complex. Notably, the lead established two significant interactions: an electrostatic interaction with GLU30:OE2, situated at a distance of 4.76631 Å, and a metal–acceptor interaction with GLU30:O, characterized by a proximity of 2.13637 Å. Cadmium’s docking within the binding sphere exhibited a more substantial CDocker energy of −30.735, denoting a robust binding interaction. The CDocker Interaction energy was calculated as −17.013, indicating a substantial interaction strength. Cadmium’s interaction was marked by a single metal–acceptor interaction with GLN21:O, occurring at an impressively close distance of 1.59521 Å, as shown in Figure 6.

## 4. Discussion

The matter of metal contamination, particularly with respect to the existence of lead and cadmium, is a vital ecological apprehension that warrants expeditious consideration. Metallothioneins (MTs) pertain to a classification of condensed proteins that are discernible by their elevated concentration of cysteine. This particular feature endows them with the ability to proficiently bind and sequester metal ions. These proteins play an indispensable role in facilitating the process of metal ion binding and sequestration. The distinctive characteristics of MTs render them highly favorable instruments for the implementation of bioremediation approaches. The principal aim of this inquiry was to scrutinize the metallothionein that was extracted from *P. putida* and assess its physicochemical characteristics, with the intent of gaining comprehension of its ability to bind lead and cadmium. The physicochemical features of metallothionein from *P. putida* were thoroughly analyzed, revealing its considerable potential for effectively binding metals and its use in bioremediation. The protein’s low molecular weight [39], presence of negatively charged residues [40,41], and hydrophilic characteristics suggest a pronounced propensity for binding positively charged metal ions, such as lead and cadmium. Moreover, based on its predicted half-life and thermostability, it may be inferred that the protein has the potential to maintain its functional properties over prolonged durations in settings contaminated by metals. The instability index, as determined by calculations, prompts inquiries regarding the protein’s stability within controlled laboratory settings. Nevertheless, it is crucial to take into account that this index may not provide a realistic representation of the protein’s behavior inside a cellular environment, as the presence of chaperones and degradation machinery could potentially impact its stability [42]. Metal and antibiotic resistance in metallothionein from several bacteria, including *P. putida*, have been reported, and therefore, it is crucial to find its virulence [52,53]. VicmPred computational investigation revealed expected functional classifications, highlighting its role in vital cellular functions and metabolic networks. A high score in the “Cellular Process” category suggests the metallothionein protein is involved in vital cellular functions. Its participation in metal ion homeostasis, stress response, and detoxification pathways suggests a role in metal binding and detoxification. The functions above demonstrate the value of this technology in bioremediation, especially where heavy metal confinement is crucial. Even though metallothionein scored poorly in “Information Molecule”, its relevance persists. Instead of transferring information, it is predicted to organize and catalyze metal ion coordination and detoxification. This is consistent with the substance’s propensity to bind heavy metals, supporting its bioremediation potential. The extraordinarily high “Metabolism” score strongly implies metallothionein’s involvement in multiple metabolic pathways. Its possible participation in metal metabolism, enzymatic activities, and metal ion transport makes it effective in lead and cadmium bioremediation. The analysis shows that metallothionein is non-pathogenic, as shown by the negative score in the “Virulence Factors” category. The above attribute is crucial when considering bioremediation purposes. The absence of virulence in this organism ensures its safe use without ecological harm.

The present in silico investigation offers significant insights regarding the appropriateness of metallothionein derived from *P. putida* for the purpose of binding lead and cadmium in bioremediation endeavors. The metallothionein from *P. putida* was analyzed using the Superfamily 1.75 tool, yielding notable results in terms of its structural classification. The designation of this molecule as a “Small protein” is based on its compact configuration, which confers benefits in terms of effective metal coordination. The categories of the “Metallothionein” fold and “Metallothionein” superfamily correspond to the principal role of the protein in binding metal ions, particularly heavy metals, such as lead and cadmium [54]. The aforementioned classification provides evidence in favor of the notion that the metallothionein derived from *P. putida* has undergone structural optimization to enhance its metal binding capabilities, hence indicating its potential utility in bioremediation endeavors. By leveraging its metal-binding properties, this protein exhibits promising potential for employment in the sequestration and immobilization of harmful metals in contaminated habitats, thereby rendering a valuable contribution to the pursuit of environmental remediation [55,56]. The exploration of post-translational modifications (PTMs) on metallothionein derived from *P. putida* via computational investigation presents a fresh perspective on the plausible regulatory mechanisms that govern its capacity to bind metals. The identification of various PTMs, including phosphorylation, glycosylation, and palmitoylation, implies the existence of a complex regulatory network that may affect the protein’s interactions with lead and cadmium ions [57]. The possibility of leveraging these PTMs presents intriguing opportunities for enhancing the metal-binding affinity of metallothionein. By selectively targeting and manipulating these modification sites, it may be feasible to design metallothionein variants that exhibit improved bioremediation capabilities. This study establishes the foundational framework for future inquiries into the functional implications of these alterations and their influence on the efficacy of metallothionein in bioremediation applications [58] and the potential functions of metallothionein from *P. putida.* The conserved domains associated with metal binding reaffirm its capacity to interact with heavy metals, such as Pb and Cd. The presented insights serve as a fundamental basis for conducting additional experimental investigations, with the goal of substantiating its metal-binding proficiency and investigating its potential for bioremediation implementations. The network’s observed PPIs underscore the intricate interdependence between proteins and pathways linked to the metallothionein from *P. putida*. The presence of 11 nodes and 15 edges suggests a multifaceted network in which the metallothionein might collaborate with other proteins to execute essential biological functions. While the PPI enrichment *p*-value of 0.0968 falls just outside the conventional significance threshold of 0.05, the trend toward enriched interactions implies a biologically relevant network that warrants further investigation [59]. The calculated average node degree of 2.73 indicates a notable degree of connectivity within the network. This level of interconnectedness might reflect a coordinated effort among proteins, possibly including metallothionein, to regulate metal homeostasis, detoxification, and other cellular processes. The relatively high average local clustering coefficient of 0.889 points to the presence of functional clusters within the network. These clusters could signify the formation of protein complexes or pathways dedicated to metal ion binding, transport, and detoxification. Identifying these clusters could guide future experimental studies aimed at deciphering the specific roles of proteins within the network [60]. The biological process annotation “Regulation of transcription, DNA-templated” is of particular interest in the context of metallothionein function. MerR Transcriptional Regulator: A significant p-value for this domain suggests a potential role in transcriptional regulation. MerR proteins are implicated in metal ion sensing and the regulation of genes involved in metal detoxification. MerR HTH Domain Type: This domain is related to the helix-turn-helix [61] motif frequently observed in transcription factors. Its presence suggests that it may participate in DNA binding and gene regulation. Superfamily of Putative DNA-Binding Domains: This superfamily is likely to comprise domains with DNA-binding properties, supporting the notion that metallothionein is involved in DNA-related processes. This annotation describes that metallothionein potentially assumes a critical function in the regulation of gene expression. This regulatory activity may potentially impact the transcription of genes that are associated with both metal detoxification and the response to environmental stressors. This potential involvement in transcriptional regulation aligns with the protein domains identified, such as the MerR transcriptional regulator and the MerR-type helix-turn-helix domain. These domains are known to be associated with metal ion sensing and gene regulation, further supporting metallothionein’s role in metal-responsive gene expression [62]. The “DNA binding” molecular function annotation strengthens the link between metallothionein and DNA-related processes. This finding suggests that metallothionein might directly interact with DNA molecules, possibly influencing the transcriptional machinery and gene regulation. The potential for DNA binding adds a layer of complexity to metallothionein’s functions, expanding its potential roles beyond metal binding alone. The presence of domains associated with DNA binding and transcriptional regulation raises intriguing possibilities for bioremediation applications. The capacity of metallothionein to effectively bind to heavy metals presents a promising opportunity to integrate its regulatory function in gene expression to develop specific bioremediation approaches. This could involve enhancing the expression of metal detoxification genes under the control of metallothionein, thereby improving the organism’s capacity to sequester and detoxify heavy metal pollutants [63]. The in silico analysis of the metallothionein sequence by the PredictProtein tool offers insights into its structural attributes that contribute to potential binding with lead and cadmium. The identified alpha-helices, beta-strands, exposed residues, flexible regions, and conserved motifs collectively suggest an intricate architecture optimized for metal binding. These structural elements likely play roles in accommodating metal ions, undergoing conformational changes, and maintaining stability [64].

Homology modeling is an indispensable technique for forecasting protein structure in the dearth of empirical evidence [65]. In this inquiry, four distinct homology modeling tools were employed to conceive models for the metallothionein sequence of *P. putida*. The variations among the models generated by PHYRE2, Robetta, ModWeb, and SwissModel can be attributed to disparities in the algorithms and databases employed by each tool. Additionally, the availability and quality of template structures may influence the content of the models [66]. The discrepancies observed in the loop regions and local structural characteristics may have implications for the metallothionein protein’s functional properties [67]. These models can be used as a starting point for additional research into the function and role of metallothionein in *P. putida*.

The validation results reveal notable differences among the computational methods in predicting the protein structure’s accuracy and reliability. Robetta stands out with the highest ProQ3 score and the most residues in the most favored region of the Ramachandran plot, indicating its superior accuracy and better agreement with known protein structures. The absence of close contacts and lower RMS deviations for bond angles and lengths further support the robustness of Robetta’s predictions. Phyre2, while demonstrating good overall performance, lags slightly behind Robetta, especially in terms of Ramachandran plot analysis and RMS deviations. However, it possesses the lowest steric clashes. ModWeb’s ProQ3 score of 0.000 raises concerns about the quality of its generated structure, potentially indicating inaccuracies or structural distortions. The positive ProsaII score and unfavorable Procheck results suggest deviations from native-like conformations. SwissModel’s performance, despite generating relatively accurate structures, falls short in certain aspects, such as the Ramachandran plot and Phi-Psi angles’ assessment. The MolProbity Clashs-core aligns with other findings, indicating the presence of steric clashes in ModWeb, SwissModel, and, to a lesser extent, Robetta. Although Robetta has exceptional performance in multiple validation criteria, Phyre2 surpasses it in terms of ProQ3 scores, Ramachandran plot conformation, and a notably lower MolProbity Clash-score. These factors establish Phyre2 as a strong competitor for the top model. The capacity of Phyre2 to produce dependable protein structures is demonstrated by its precise modeling of the complete length of residues and its effective avoidance of steric conflicts.

The metallothionein from *P. putida*, as revealed by Superfamily 1.75 analysis, exhibits a compact size that is consistent with its categorization as a “Small protein”. This size confers unique advantages, allowing the protein to efficiently coordinate metal ions within its confined structural environment. Additionally, the categorization of the protein under the “Metallothionein” fold and “Metallothionein” superfamily underscores its central role in binding and sequestering metal ions [68]. This structural motif is characterized by conserved cysteine residues crucial for metal coordination, further reinforcing its suitability for metal binding and detoxification [69]. The docking results, in relation to the domain architecture and conserved residues, establish a clear link between the observed interactions and the protein’s molecular attributes. The lead binding simulation, characterized by a negative CDocker energy of −0.10587, aligns seamlessly with the protein’s inherent negatively charged residues and hydrophilic character. These properties create an environment conducive to interactions with positively charged metal ions, such as Pb. Notably, the CDocker Interaction energy of −0.10587 signifies the stability of the Pb-MT complex, a testament to the protein’s structural features that facilitate the binding process. Importantly, the docking interactions involving lead resonate with the conserved residues identified in the protein sequence. The electrostatic interaction with GLU30:OE2 at 4.76631 Å and the metal–acceptor interaction with GLU30:O at 2.13637 Å underscore the strategic role of GLU30 in mediating metal binding. These interactions substantiate the predictions based on the protein’s physicochemical properties, highlighting GLU30’s versatility in coordinating metal ions. Similarly, the cadmium binding simulation, characterized by a more substantial CDocker energy of −30.735 and an interaction energy of −17.013, aligns seamlessly with the protein’s attributes conducive to metal binding. The ability to maintain its structure and stability across diverse conditions further underscores its suitability for robust metal coordination. The interaction between cadmium and the specific residue GLN21:O at an impressively close distance of 1.59521 Å reaffirms the role of this residue in cadmium binding. These docking interactions seamlessly correlate with the earlier analyses of the protein’s conserved residues, highlighting the functional significance of specific amino acids in facilitating metal coordination.

## 5. Conclusions

In conclusion, this study illuminates the promising potential of metallothionein derived from *P. putida* in combatting the critical issue of heavy metal contamination. Through an extensive analysis that encompassed physicochemical exploration, structural evaluation, molecular docking, and scrutiny of protein–protein interactions, a robust basis has been established to support its application in bioremediation. The protein’s compact structure, enriched cysteine composition, and strategic arrangement of negatively charged residues synergistically equip it with a remarkable capacity to bind and sequester heavy metal ions, with particular emphasis on lead and cadmium. Key interacting residues come to the forefront in molecular docking simulations: the electrostatic association involving GLU30:OE2 at 4.76631 Å, along with the metal–acceptor interaction featuring GLU30:O at 2.13637 Å, distinctly underscore the centrality of GLU30 in steering lead binding. Similarly, the cadmium docking interactions highlight the indispensable contribution of GLN21:O at an impressive proximity of 1.59521 Å, thus underscoring the pivotal role of this residue in cadmium coordination. Furthermore, insight into potential avenues for enhancing metal-binding capacity is offered by post-translational modification analysis, opening doors for targeted engineering. The intricate network of protein–protein interactions accentuate its involvement in critical cellular processes, amplifying its potential roles in detoxification pathways. This study bridges theoretical analyses with practical applications, paving the way for future research, experimental endeavors, and innovative bioremediation techniques. Metallothionein derived from *P. putida* arises as a promising candidate for environmental remediation, poised to exert a significant influence in mitigating the negative effects of heavy metal pollution.

## Figures and Tables

**Figure 1 toxics-11-00864-f001:**
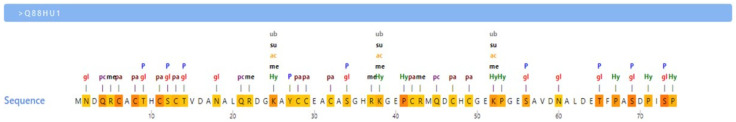
Post-translational modification sites in metallothionein. Abbreviations: P: phosphorylation; gl: glycosylation; ub: ubiquitination; su: SUMOylation; ac: acetylation; me: methylation; pc: pyrrolidone carboxylic acid; pa: palmitoylation; Hy: hydroxylation.

**Figure 2 toxics-11-00864-f002:**
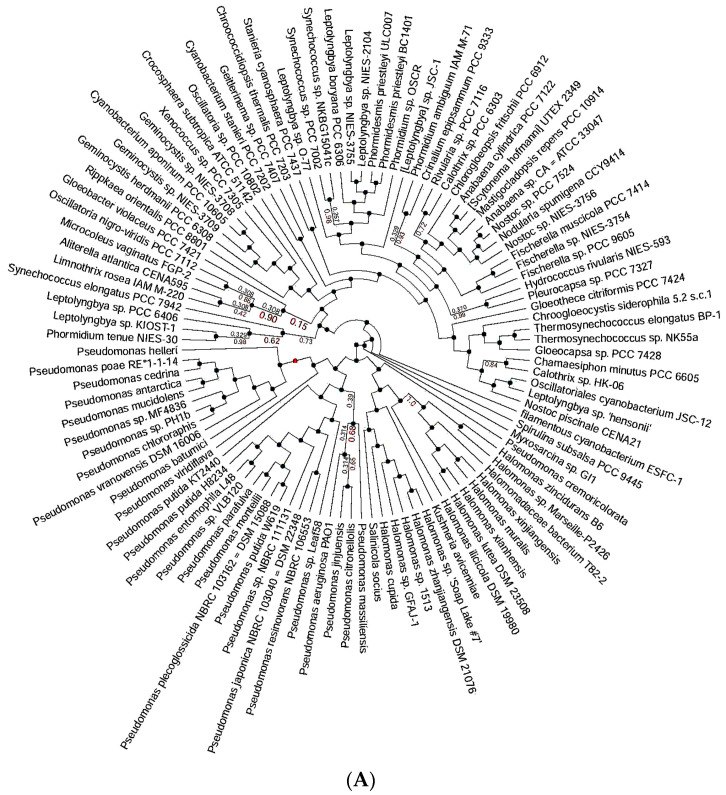
EggNOG analysis of the metallothionein from *P. putida* exploring the orthologs. (**A**) Representation of all orthologs from various species, (**B**) highlighted orthologs from *Pseudomonas* species.

**Figure 3 toxics-11-00864-f003:**
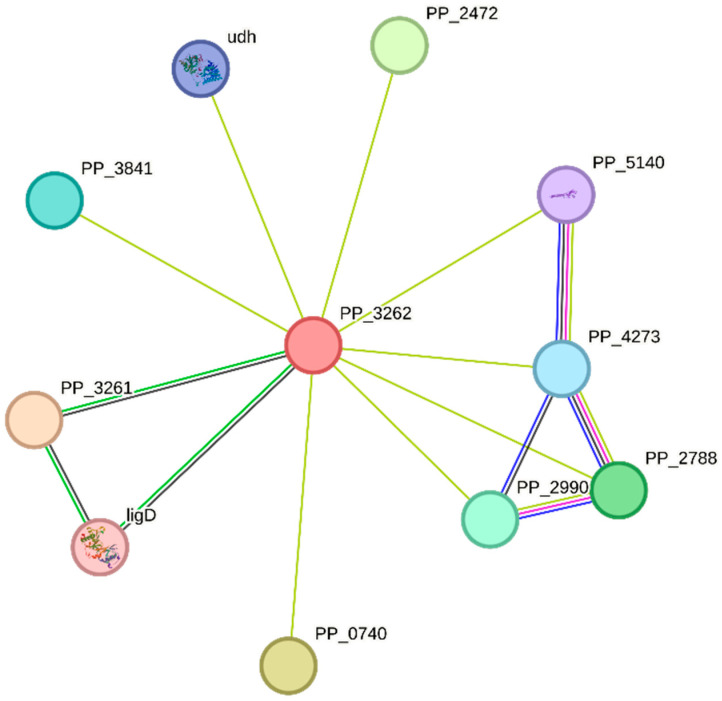
Protein–protein interaction network of *P. putida* metallothionein (PP_3262). Note: The red-colored node in the center represents the query protein, metallothionein (MT). Nodes with a filled structure indicate the presence of a 3D structure of the respective proteins. Conversely, empty nodes denote proteins for which 3D structures are unknown. The edges connecting the nodes represent different types of protein–protein associations: Green edges indicate gene neighborhood relationships; pink edges represent known interactions determined experimentally; blue edges signify predicted gene co-occurrence associations; yellow edges denote text-mined interactions; brown edges represent predicted co-expression relationships. This comprehensive visualization provides insights into the relationships between metallothionein and other proteins in the context of various association types.

**Figure 4 toxics-11-00864-f004:**
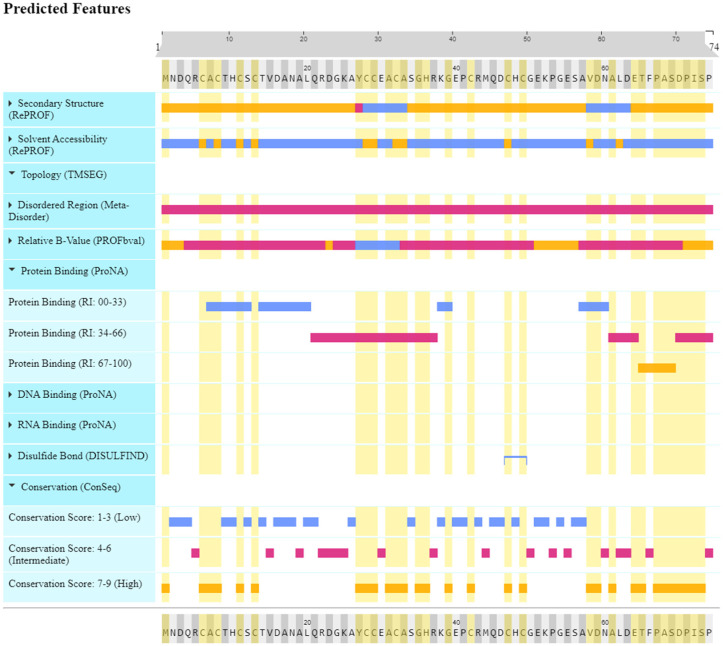
Metallothionein sequence analysis showing post-translational modifications and conserved residues.

**Figure 5 toxics-11-00864-f005:**
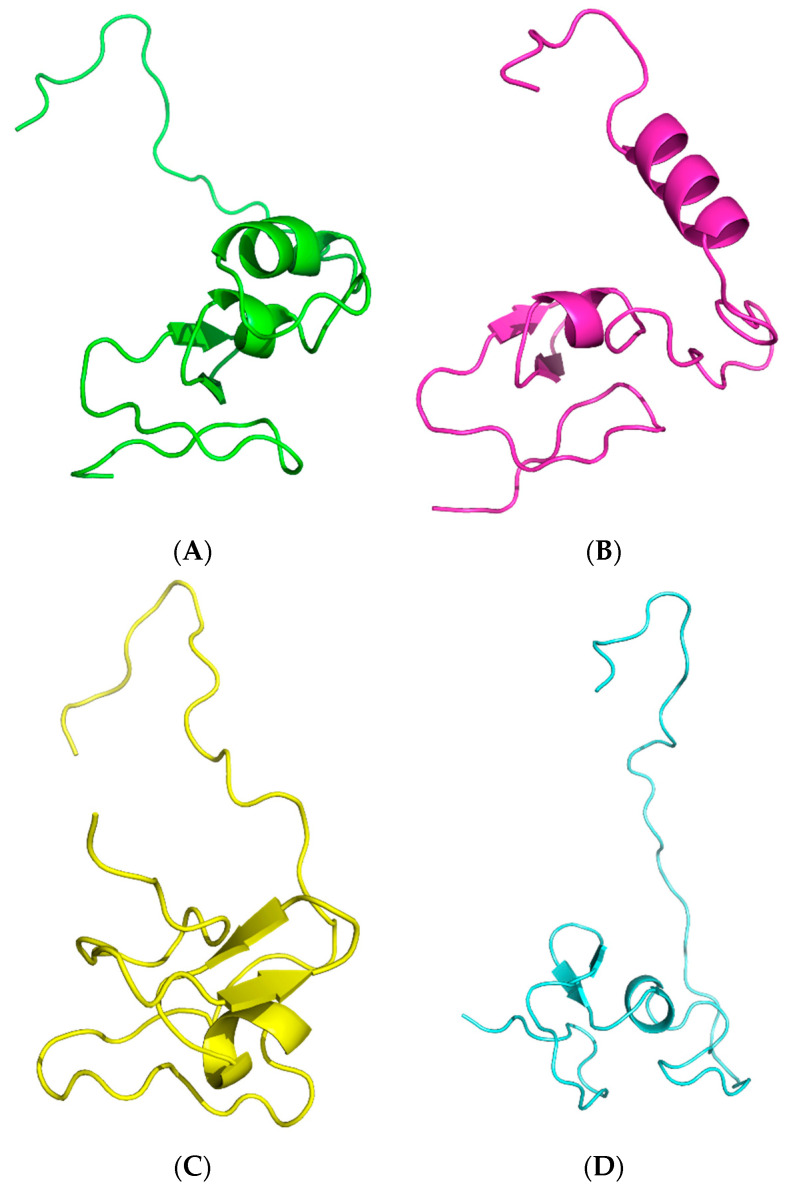
Homology-modeled structures of *P. putida* metallothionein shown in cartoon representation models modeled by: (**A**) Phyre2, (**B**) Robetta, (**C**) ModWeb, and (**D**) SwissModel.

**Figure 6 toxics-11-00864-f006:**
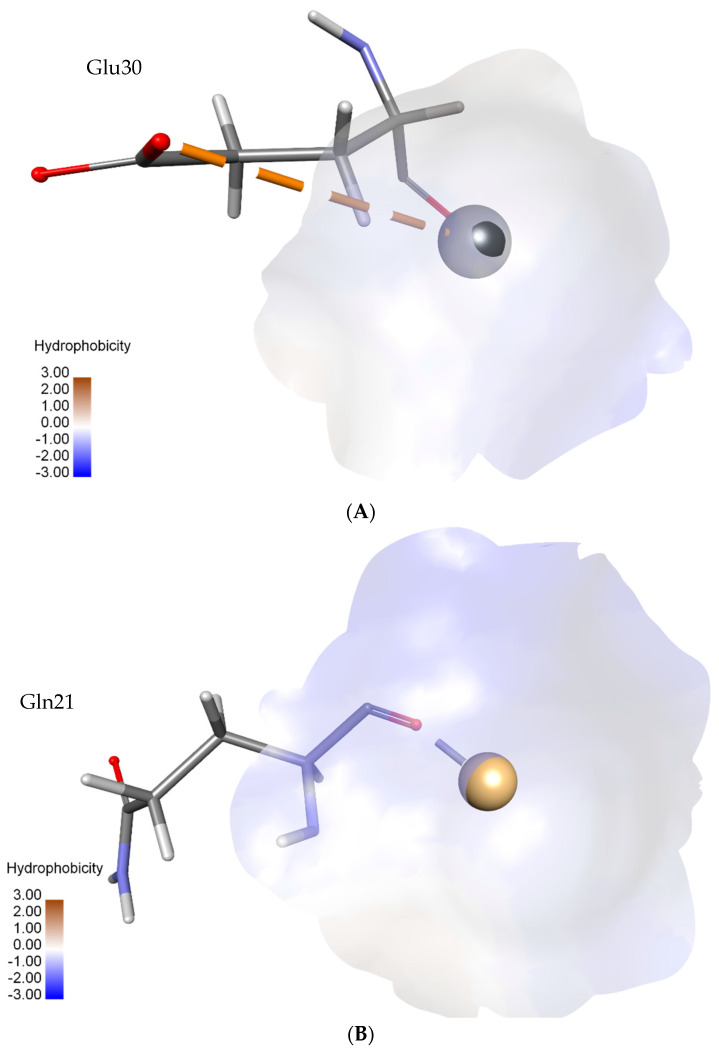
Molecular docking was performed to understand the interactions between metallothionein from *P. putida* and heavy metals: lead and cadmium. (**A**) The docking model illustrates the interaction between metallothionein and lead ion (in gray), showcasing the crucial role of GLU30:OE2 and GLU30:O residues in coordinating lead binding. (**B**) The docking model demonstrates the interaction between metallothionein and cadmium ion (in orange), emphasizing the significant contribution of GLN21:O residue in facilitating cadmium binding.

**Table 1 toxics-11-00864-t001:** Validation results of protein structure prediction methods.

Model Validation Tool	Phyre2	Robetta	ModWeb	SwissModel
Residues built	1–74	1–74	1–73	1–73
ProQ3	0.383	0.467	0.423	0.000
**Ramachandran Plot Summary**				
Most favored	84.1%	95.4%	87.1%	75.8%
Additionally allowed	12.7%	4.6%	11.3%	21.0%
Generously allowed	3.2%	0.0%	1.6%	3.2%
Disallowed	0.0%	0.0%	0.0%	0.0%
**Close Contacts and Deviations from Ideal Geometry**				
Number of close contacts (within 2.2 Å)	0	0	0	0
RMS deviation for bond angles	1.9°	2.2°	2.3°	2.2°
RMS deviation for bond lengths	0.020 Å	0.016 Å	0.020 Å	0.016 Å
**Global quality scores**				
Verify3D	−6.10	−5.78	−6.90	−6.74
ProsaII (-ve)	−0.37	−0.37	1.03	−0.79
Procheck (phi-psi) ^3^	−0.43	0.51	−0.75	−3.03
Procheck (all) ^3^	0.24	1.18	−0.59	−2.90
MolProbity Clashscore	−19.42	1.09	1.53	1.36

## Data Availability

The research reported in the article used no data.

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
