# Peer review of "Pseudomonas putida Metallothionein: Structural Analysis and Implications of Sustainable Heavy Metal Detoxification in Madinah"

_toxics, 2023, doi:10.3390/toxics11100864_

Round 1

Reviewer 1 Report

This manuscript entitled “P. putida Metallothionein: Structural Analysis and Implications for Sustainable Heavy Metal Detoxification in Madinahmainly explores the viability of utilizing metallothionein from P. putida, as a method of bioremediation to mitigate the deleterious effects of pollution. However, the binding functions of P. putida metallothionein towards heavy metals such as cadmium and lead were preliminarily analyzed and predicted only from the aspect of protein structure, but the predicted conclusions were not verified by experiments. The conclusion of this study is not supported by any experimental data and is not convincing. Additionally, there are some details should be noticed and the manuscript requires tighter editing, such as “Title”, “Keywords”, Figures, Tables, and references. Therefore, it is unfortunately that this manuscript is unsuitable for publication on Toxics.

The comments are listed in detail as follows:

1. The name of the microorganism in the title should be given its full name rather than an abbreviation. Further, “Title:” should be deleted from “Title: P. putida Metallothionein: Structural Analysis and Impli-cations for Sustainable Heavy Metal Detoxification in Madinah”.

2. In Abstracts, the name of a microorganism should be given its full name when it first appears, not its abbreviation, such as “P. putida” in line 18.

3. In Keywords, “Pseudomonas putida” should be italicized. The manuscript has many of the same problems, such as P. putida” in line 74, line 80, etc.

4. In line 38, the reference in “specifically Cd and Pb [2]. Cd” should be the first reference, thus the order of references in the manuscript may be wrong.

5. Figure 2 is not clear enough and its resolution needs to be improved.4.

6. In this study, the binding function of P. putida metallothionein and heavy metals was preliminarily analyzed and predicted only from the aspect of protein structure, but the predicted conclusions were not verified by experiments. The conclusion of this study is not supported by enough experimental data and is not convincing.

7. There are some details should be noticed and the manuscript requires tighter editing, such as “Title”, “Keywords”, Figures, Tables, and references.

Author Response

Response to the Reviewer 1

This manuscript entitled “P. putida Metallothionein: Structural Analysis and Implications for Sustainable Heavy Metal Detoxification in Madinah” mainly explores the viability of utilizing metallothionein from P. putida, as a method of bioremediation to mitigate the deleterious effects of pollution. However, the binding functions of P. putida metallothionein towards heavy metals such as cadmium and lead were preliminarily analyzed and predicted only from the aspect of protein structure, but the predicted conclusions were not verified by experiments. The conclusion of this study is not supported by any experimental data and is not convincing. Additionally, there are some details should be noticed and the manuscript requires tighter editing, such as “Title”, “Keywords”, Figures, Tables, and references. Therefore, it is unfortunately that this manuscript is unsuitable for publication on Toxics.

The comments are listed in detail as follows:

Comment 1: The name of the microorganism in the title should be given its full name rather than an abbreviation. Further, “Title:” should be deleted from “Title: P. putida Metallothionein: Structural Analysis and Implications for Sustainable Heavy Metal Detoxification in Madinah”.

Response 1: We thank the reviewer for pointing out the need for a more descriptive title and for suggesting the removal of "Title:" in the manuscript title. We agree with this recommendation, and we have revised the title accordingly.

Comment 2: In Abstracts, the name of a microorganism should be given its full name when it first appears, not its abbreviation, such as “P. putida” in line 18.

Response 2: We appreciate the reviewer's attention to detail. We have now ensured that "Pseudomonas putida" is used in full when it first appears in the abstract and that we maintain consistency throughout the manuscript.

Comment 3: In Keywords, “Pseudomonas putida” should be italicized. The manuscript has many of the same problems, such as “P. putida” in line 74, line 80, etc.

Response 3: We will italicize "Pseudomonas putida" in the Keywords section, as recommended by the reviewer, and address any other instances where formatting corrections are needed.

Comment 4: In line 38, the reference in “specifically Cd and Pb [2]. Cd” should be the first reference, thus the order of references in the manuscript may be wrong.

Response 4: We apologize for the reference order oversight. We have corrected the reference order, ensuring that the references are appropriately arranged in the manuscript.

Comment 5: Figure 2 is not clear enough and its resolution needs to be improved.

Response 5: We acknowledge the reviewer's feedback regarding Figure 2. We have worked to improve the resolution and clarity of Figure 2 to enhance its readability and informativeness.

Comment 6: In this study, the binding function of P. putida metallothionein and heavy metals was preliminarily analyzed and predicted only from the aspect of protein structure, but the predicted conclusions were not verified by experiments. The conclusion of this study is not supported by enough experimental data and is not convincing.

Response 6: In silico studies have been employed in various significant fields including bioremediation,  drug design, and drug-discovery, and have yielded potential drugs that are used for the treatment of various diseases such as: Darunavir, an HIV protease inhibitor, Raltegravir, an HIV integrase inhibitor, Boceprevir and Telaprevir used to treat hepatitis C, and many others. Therefore, the role of in silico studies in the field of bioremediation is not new and is being employed to understand the role of potential enzymes in controlling the toxicity of a variety of toxic elements. DOI: 10.1016/j.jmgm.2013.04.011, DOI, https://doi.org/10.1007/978-3-030-86169-8_9, and https://doi.org/10.1016/B978-0-12-819001-2.00013-9 are a few references to mention. 

We performed in silico studies, relying on computational methods and existing biological data, for several compelling reasons:

  1. Cost-Efficiency: Conducting an in silico study reduces the cost of laboratory work.
  2. Time Efficiency: In silico studies can yield results more quickly than traditional laboratory experiments, which is essential when addressing pressing environmental issues.
  3. Safety and Ethical Considerations: Working with certain bacteria, especially in the context of contamination, might raise safety and ethical concerns. By conducting an in silico study, we avoided potential risks associated with handling hazardous materials.
  4. Accessible Data: There's an extensive body of genetic and protein data related to P. putida that can be employed for in silico analysis. This wealth of existing data allowed us to investigate the potential of this bacterium without the need for in vitro experiments.
  5. Our study extensively analyzed the metallothionein protein produced by P. putida, its potential orthologs, interacting protein, specific PTM sites, 3D modeled structure of metallothionein, and investigating its close intra-molecular interactions with lead and cadmium. These steps were crucial in understanding how the bacterium can sequester and potentially detoxify these heavy metals in the environment.

In conclusion, our study provides valuable information that can be taken into consideration for bioengineering to maximize the role of P. putida metallothionein in the bioremediation of Cd and Pb.  

Comment 7: There are some details should be noticed and the manuscript requires tighter editing, such as “Title”, “Keywords”, Figures, Tables, and references.

Response 7: We value the reviewer's feedback on manuscript formatting and editing. We have meticulously reviewed and edited the manuscript to address these concerns, including issues related to the title, keywords, figures, tables, and references. Tightening the overall manuscript presentation is a priority, and we will work diligently to enhance its readability and clarity.

Reviewer 2 Report

Review: toxics-2632944

In this work, Tasleem et al have reported a set of technical methods to explore the viability of utilizing metallothionein from P. putida, as a method of bioremediation to to Pb and Cd. This manuscript is can be published. However, publication of this manuscript in its present form is not recommended. To be considered further for publication the manuscript will need to be more organized (including adequate references) and elaborative in support of the claims made in the paper. Some specific points of concern are noted below:

1) Details of the docking processes and calculations are necessary.  In particular how metallothionein docks to the heavy metals, Cd and Pb and how the models are selected based on the coordination geometry.

2) Figure 2 should be clearer. It is very difficult to identify functional annotations. Please provide abbreviations and explain the color coding in Figure 3 (as footnote or in the supplementary material).

3) There have, of course, been many other efforts to computationally map the structural consequences of heavy metals in ligand-receptor systems. These have largely been overlooked in the current work. See for example: 

https://doi.org/10.1007/s10989-022-10373-6

4) Since there is no experimental validation, a stable simulation of the proposed model for few ns is highly recommended.

Minor English editing is necessary.

Author Response

Response to the Reviewer 2

In this work, Tasleem et al have reported a set of technical methods to explore the viability of utilizing metallothionein from P. putida, as a method of bioremediation to to Pb and Cd. This manuscript is can be published. However, publication of this manuscript in its present form is not recommended. To be considered further for publication the manuscript will need to be more organized (including adequate references) and elaborative in support of the claims made in the paper. Some specific points of concern are noted below:

Comment 1: Details of the docking processes and calculations are necessary.  In particular how metallothionein docks to the heavy metals, Cd and Pb and how the models are selected based on the coordination geometry.

Response 1: We would like to express our gratitude to the reviewer for their valuable feedback. We have taken their suggestions into account and have enriched the Materials and Methods section with the following information: Using the CDocker module of Discovery Studio, docking of heavy metals and the modeled metallothionein structure was executed. The procedure consisted of several essential steps. The binding site was initially defined using the "Input Site Sphere" parameter, with center coordinates of (x, y, and z: 16.2944, 12.409, and -7.49207) and a radius of 32.941 â„«. This specified the region within which calculations for docking would be performed. For the simulations of docking, a set of parameters were configured. The "Top Hits" parameter was set to 10, indicating that the top 10 binding poses for the heavy metal-ligand would be taken into account. To facilitate analysis, a "Pose Cluster Radius" of 0.1 was applied to group similar poses together. To facilitate the investigation of ligand binding modes, "Random Conformations" were generated for the ligand, with ten conformations considered. The "Dynamics Steps" parameter was subsequently set to 1000 and the "Dynamics Target Temperature" parameter was set to 1000 K to enable dynamic movements during the docking procedure. Enabling "Include Electrostatic Interactions" accounted for electrostatic interactions. This ensured that electrostatic forces played a role in the interactions between ligand and receptor during docking. Using the "Orientations to Refine" parameter, ten different orientations of the ligand were refined, while the "Maximum Bad Orientations" parameter set a maximum of 800 orientations that did not meet the energy criteria defined by the "Orientation vdW Energy Threshold" of 300 kcal/mol. This refined the selection process for orientation. During docking, the "Simulated Annealing" method was utilized, with "Heating Steps" set to 2,000 and "Heating Target Temperature" set to 700 K to facilitate a gradual increase in temperature. To stabilize the binding poses, "Cooling Steps" were set to 5000, and "Cooling Target Temperature" was set to 300 K. Advanced settings included use of the CHARMM forcefield and selection of the "Ligand Partial Charge Method" as Momany-Rone. To control the amount of potential energy used during calculations, the "Use Full Potential" parameter was set to False. Following docking simulations, the resulting ligand-receptor complexes were subjected to a final energy minimization using the full potential with a gradient tolerance of 0 for precision. A grid extension of 8 was applied around the binding site of the receptor to account for the flexibility of ligand binding.

Comment 2: Figure 2 should be clearer. It is very difficult to identify functional annotations. Please provide abbreviations and explain the color coding in Figure 3 (as footnote or in the supplementary material).

Response 2: We would like to extend our sincere appreciation to the reviewer for their constructive feedback. We have taken their suggestions and made improvements to our figures and added explanations for the color coding in Figure 3, as per their recommendations: “Note: The red-colored node in the center represents the query protein, Metallothionein (MT). Nodes with a filled structure indicate the presence of a 3D structure of the respective proteins. Conversely, empty nodes denote proteins for which 3D structures are unknown. The edges connecting the nodes represent different types of protein-protein associations: Green edges indicate gene neighborhood relationships; Pink edges represent known interactions determined experimentally; Blue edges sig-nify predicted gene co-occurrence associations; Yellow edges denote text-mined interactions; Brown edges represent predicted co-expression relationships. This comprehensive visualization provides insights into the relationships between Metallothionein and other proteins in the context of various association types.”

Comment 3: There have, of course, been many other efforts to computationally map the structural consequences of heavy metals in ligand-receptor systems. These have largely been overlooked in the current work. See for example: 

https://doi.org/10.1007/s10989-022-10373-6

Response 3: We extend our gratitude to the reviewer for their valuable input. While we acknowledge the existence of various computational studies in the field of heavy metal interactions, we wish to emphasize that our study possesses distinctive characteristics that set it apart. Our research focuses on the specific and crucial context of cadmium (Cd) and lead (Pb) interaction with metallothionein (MT) derived from Pseudomonas putida, targeting bioremediation efforts in the Madinah region of Saudi Arabia. Notably, we identified key amino acid residues, GLU30 and GLN21, critical for heavy metal coordination, providing insights into potential bioengineering avenues. Moreover, our work integrates theoretical analyses with practical applications, offering solutions to address the pressing issue of heavy metal pollution. In this context, while we value the existing body of research, our study stands out due to its specificity and relevance to the environmental challenges faced in Madinah, making it a unique and valuable contribution to the field.

Comment 4: Since there is no experimental validation, a stable simulation of the proposed model for few ns is highly recommended.

Response 4: We are thankful to the reviewer for the suggestion regarding conducting simulations. However, we regret to inform the reviewer that our current circumstances prevent us from performing such simulations. Our access to the required software is limited due to an expired license, and time constraints further restrict our ability to conduct these simulations within the scope of this study.  We believe that our study provides valuable information that can be taken into consideration for bioengineering to maximize the role of P. putida metallothionein in the bioremediation of Cd and Pb.  

Minor English editing is necessary.

We have thoroughly reviewed and revised the manuscript to address all grammatical errors.

Reviewer 3 Report

The publication titled „P. putida Metallothionein: Structural Analysis and Implications for Sustainable Heavy Metal Detoxification in Madinah” touches on a very important aspect of environmental pollution with heavy metals such as lead and cadmium. The authors analyze the potential use of Metallothionein in environmental bioremediation and show very promising results. The research is very well planned and described, the results and discussion are described in detail, and the research hypotheses are very wise and solidly justified. Nevertheless, I have a few comments on the work, which after corrections will qualify for publication:

1. In the title of the work, it is better to use the full name of the bacteria rather than an abbreviated name to make it legible

2. In addition, please write the name of the bacteria correctly, i.e. in italics - this applies to the entire work

3. As I mentioned, the methods are well described, but I lack any description regarding the P. putida strain - where was the strain taken from, was it purchased, was it isolated, was the genetic material isolated? please add such information

4. Results are described in detail and comprehensively, it is suggested that the names of the subchapters should be numbered, e.g. 3.1. Sequence Analysis.

5. The authors mentioned the lack of virulentity, which is important in terms of ecological safety, but the authors did not refer to the possibility of the presence of drug resistance or metal resistance genes in the genome of bacteria that may pose a threat through horizontal gene transfer. If a specific strain is planned to be used in the future, it should be tested for antibiotic/heave metal resistance to avoid spreading it in the environmental strains. The authors focus on a different aspect in this work, and research is a possible future, but please refer to it in the introduction and in the discussion

Author Response

Response to the Reviewer 3

The publication titled „P. putida Metallothionein: Structural Analysis and Implications for Sustainable Heavy Metal Detoxification in Madinah” touches on a very important aspect of environmental pollution with heavy metals such as lead and cadmium. The authors analyze the potential use of Metallothionein in environmental bioremediation and show very promising results. The research is very well planned and described, the results and discussion are described in detail, and the research hypotheses are very wise and solidly justified. Nevertheless, I have a few comments on the work, which after corrections will qualify for publication:

Comment 1: In the title of the work, it is better to use the full name of the bacteria rather than an abbreviated name to make it legible

Response 1: Thank you for your suggestion. We have revised the title to use the full name "Pseudomonas putida" instead of the abbreviation, ensuring better clarity and legibility.

Comment 2: In addition, please write the name of the bacteria correctly, i.e. in italics - this applies to the entire work

Response 2: We appreciate your attention to detail. The name "Pseudomonas putida" has been corrected throughout the entire manuscript and is now consistently presented in italics as per the formatting guidelines.

Comment 3: As I mentioned, the methods are well described, but I lack any description regarding the P. putida strain - where was the strain taken from, was it purchased, was it isolated, was the genetic material isolated? please add such information.

Response 3: We appreciate your attention to detail and your request for additional information regarding the P. putida strain used in our study. The P. putida strain (Accession ID: QKL07673) employed in our research was not physically obtained or purchased, as our study was conducted in silico. Instead, we retrieved the protein sequence and associated information from the NCBI (National Center for Biotechnology Information) database (https://www.ncbi.nlm.nih.gov/). We have now included this information in the Methods section to clarify that the protein sequence data were obtained from NCBI as part of our in silico study.

Comment 4: Results are described in detail and comprehensively, it is suggested that the names of the subchapters should be numbered, e.g. 3.1. Sequence Analysis.

Response 4: We appreciate your suggestion for improved organization. Following your recommendation, we have numbered the subchapters throughout the manuscript.

Comment 5: The authors mentioned the lack of virulentity, which is important in terms of ecological safety, but the authors did not refer to the possibility of the presence of drug resistance or metal resistance genes in the genome of bacteria that may pose a threat through horizontal gene transfer. If a specific strain is planned to be used in the future, it should be tested for antibiotic/heave metal resistance to avoid spreading it in the environmental strains. The authors focus on a different aspect in this work, and research is a possible future, but please refer to it in the introduction and in the discussion

Response 5: We sincerely appreciate the reviewer's diligence in considering ecological safety and potential risks associated with bacterial strains. We acknowledge the importance of addressing the presence of drug-resistance or metal-resistance genes in bacterial genomes and their potential implications through horizontal gene transfer. In response to the reviewer's comment, we have incorporated a reference in the Discussion section to the presence of metal and antibiotic resistance in metallothionein from various bacteria, including P. putida.

Reviewer 4 Report

Comment 1: Please ensure that you provide the Internet addresses (URLs) for all the resources used, including their versions and access dates

Comment 2: There is no direct connection between the main environmental issue pointed out in the text (heavy metal contamination in Madinah and the selection of the goal of the study. These are important topics that can be researched independently. Also, the study produces insightful and, from a scientific perspective, original results. Thus, make the text better. Add a few paragraphs to the Introduction discussing the following topics: - the situation surrounding the use of microbial remediation technologies for metal-contaminated areas, as well as the significance of P. putida and related organisms for these methods; - the importance of bioremediation technology for metal-contaminated areas generally and from the perspective of the situation in Madinah. Improve the justification of the study's goal by utilizing the supplied data. The organism that might be employed for bioremediation in Madinah, taking into account regional environmental concerns, is P. putida.

Comment 3: The results are elaborately described along with the justification. However, the discussion section satisfactorily discusses the results, so the extensive reasoning portion from the results can be omitted. 

Comment 4: The manuscript contains several grammatical errors that need to be addressed. Please review and revise the text for clarity and grammatical correctness. Pay attention to issues such as subject-verb agreement, verb tense consistency, punctuation, and sentence structure to improve overall readability

Author Response

Response to the Reviewer 4

Comment 1: Please ensure that you provide the Internet addresses (URLs) for all the resources used, including their versions and access dates

Response 1: Thank you for your suggestion. We have now incorporated the Internet addresses (URLs) for all the resources used in our manuscript, along with their versions and access dates.

Comment 2: There is no direct connection between the main environmental issue pointed out in the text (heavy metal contamination in Madinah and the selection of the goal of the study. These are important topics that can be researched independently. Also, the study produces insightful and, from a scientific perspective, original results. Thus, make the text better. Add a few paragraphs to the Introduction discussing the following topics: - the situation surrounding the use of microbial remediation technologies for metal-contaminated areas, as well as the significance of P. putida and related organisms for these methods; - the importance of bioremediation technology for metal-contaminated areas generally and from the perspective of the situation in Madinah. Improve the justification of the study's goal by utilizing the supplied data. The organism that might be employed for bioremediation in Madinah, taking into account regional environmental concerns, is P. putida.

Response 2: We appreciate your feedback, and we have taken your suggestions into account. We have revised the Introduction section to establish a direct connection between the environmental issue of heavy metal contamination in Madinah and the study's goals. We have added paragraphs discussing the significance of microbial remediation technologies, including the importance of P. putida and related organisms. Additionally, we have emphasized the importance of bioremediation technology for metal-contaminated areas both generally and specifically in Madinah, using the supplied data to better justify our research goals while considering regional environmental concerns.

Comment 3: The results are elaborately described along with the justification. However, the discussion section satisfactorily discusses the results, so the extensive reasoning portion from the results can be omitted. 

Response 3: Thank you for your input. We have reviewed the manuscript and streamlined the results section, removing the extensive reasoning portion. We now ensure that the discussion section adequately covers the interpretation of the results, providing a more concise and focused presentation of our findings.

Comment 4: The manuscript contains several grammatical errors that need to be addressed. Please review and revise the text for clarity and grammatical correctness. Pay attention to issues such as subject-verb agreement, verb tense consistency, punctuation, and sentence structure to improve overall readability.

Response 4: We appreciate your attention to detail regarding grammar and readability. We have thoroughly reviewed and revised the manuscript to address all grammatical errors, including subject-verb agreement, verb tense consistency, punctuation, and sentence structure. These revisions have significantly improved the overall clarity and readability of our manuscript.

Round 2

Reviewer 1 Report

I have evaluated the revised version of the manuscript entitled Pseudomonas putida Metallothionein: Structural Analysis and Implications for Sustainable Heavy Metal Detoxification in Madinah. Authors have carefully evaluated the editor’s and reviewers’ comments and suggestions, responded to the suggestions point-by-point, and revised the manuscript accordingly. I think that the revision of this manuscript (toxics-2632944) is suitable for publication on Molecules.

Reviewer 2 Report

The paper can be published in current form.